# Intra and Inter-Observer Reliability and Repeatability of Metatarsus Adductus Angle in Recreational Football Players: A Concordance Study

**DOI:** 10.3390/jcm11072043

**Published:** 2022-04-06

**Authors:** Eduardo Pérez Boal, Carlos Martin-Villa, Ricardo Becerro de Bengoa Vallejo, Marta Elena Losa Iglesias, Bibiana Trevissón Redondo, Israel Casado Hernández, César Calvo Lobo, David Rodríguez Sanz

**Affiliations:** 1Facultad de Enfermería y Fisioterapia, Universidad de León, 24401 Ponferrada, Spain; epereb@unileon.es (E.P.B.); btrer@unileon.es (B.T.R.); 2Facultad de Enfermería, Fisioterapia y Podología, Universidad Complutense de Madrid, 28040 Madrid, Spain; ribebeva@ucm.es (R.B.d.B.V.); isracasa@ucm.es (I.C.H.); cescalvo@ucm.es (C.C.L.); davidrodriguezsanz@ucm.es (D.R.S.); 3Facultad de Ciencias de la Salud, Universidad Rey Juan Carlos, 28933 Madrid, Spain; marta.losa@urjc.es

**Keywords:** metatarsus adductus, sports, football, radiology, musculoskeletal disease

## Abstract

Metatarsus adductus (MA) is a congenital foot deformity often unrecognized at birth. There is adduction of the metatarsals, supination of the subtalar joint, and plantarflexion of the first ray. The aims of this study were to assess the intra and inter-reader reliability of the radiographic MA measurement angles used in the literature. Methods: All consecutive recreational football players who practice activity more than 5 h/week over 21 years of age with MA by roentgenographic evaluation on weight-bearing dorsoplantar images were included in a cross-sectional study. Controls were matched to cases according to age and gender. We assess all radiographic measurements to evaluate metatarsus adductus with the different measurements frequently used in the literature: Sgarlato, modified Sgarlato, Rearfoot, Root, Engel, modified Engel, Kite, Kilmartin, modified Kilmartin, Simons, and Laaveg & Ponseti. Results: The variables measured in 80 weight-bearing dorsoplantar foot radiographs show excellent reliability ranging *p* > 0.900 in Sgarlato and modified Sgarlato with low SEM, CV, and MCD. Rearfoot, Root, Engel, modified Engel, Kite, Kilmartin, Simons, Laaveg & Ponseti, and modified Kilmartin’s angles showed intra or inter reliability with ICC lower than <0.900, systematic differences between intersession or inter observers, or high MCD value. Conclusion: It is more suitable to measure the MA angle with the Sgarlato and modified Sgarlato techniques to show higher reliability and repeatability for intra and inter-observer.

## 1. Introduction

Metatarsus adductus (MA) is a relatively common congenital foot deformity that is often unrecognized at birth [1]. There is adduction of the metatarsals, supination of the subtalar joint, and plantarflexion of the first ray [2]. Metatarsus adductus is defined as a transverse plane deformity in which the metatarsals deviate medially in relation to the midfoot. This can present as one of the deformities associated with clubfeet in the paediatric population, or be observed in adolescents and adults [3,4,5,6]. The frequency of occurrence of metatarsus adductus varies from 0.1% to 12% [7,8,9]. This pathology is not completely clear [10,11]. The exact aetiology of metatarsus adductus is unknown, though it has been suggested that increased intrauterine pressure, osseous abnormality, and abnormal muscle attachments may be causes [12].

A growing body of evidence has investigated the reliability of various angular and linear measurements on X-rays of the foot and ankle. The reliability of these measurements has been shown to vary for different methods used. However, higher inter-observer than intra-observer disagreement is commonplace [13,14,15]. This may be explained by the lack of unanimity of landmarks used in charting different angular measurements [16].

Reliability is a fundamental problem for measurement in all of science [17]. The aims of the present study were to assess intra and inter-reader repeatability and reliability and obtain the mean values of the metatarsus adductus angle in men and women by using the measurement system most used in the literature [18] in order to find the most suitable measurement for valuation of MA. Because few studies compare intra and inter-observer measurements, we want to present the most effective measures for a complete assessment of MA. We consider this study to be very important in helping physicians become highly efficient when carrying out radiological evaluations.

No previous publications in review evaluate all the methods currently used to measure the metatarsus adductus deformity. As a main objective, we have developed a study including the measurements currently described and evaluating their reliability and repeatability.

## 2. Materials and Methods

### 2.1. Study Design

A reliability study was carried out to determine the intra and inter-rater reliability of 11 different measurements of MA in two sessions.

### 2.2. Sample Size Calculation

The required sample size was calculated based on reliability testing to determine reliability. In this study, the ICCs were used for reliability testing at a target value of 0.8 and a 95% CI of 0.2. We calculated the sample size to be 36 patients with Bonett’s approximation [19].

### 2.3. Ethical Aspects

This research was approved by the local Research Ethical Committee at the University of Rey Juan Carlos (URJC) in Madrid, with internal register number 0212201600117. The required local regulations and ethical standards for human experimentation of the Declaration of Helsinki were respected [20].

### 2.4. Subjects

The sample was recreational football players who practice more than 5 h/week, attending the foot and ankle clinical unit at the CEMTRO hospital of Madrid from September 2019 to February 2020. All consecutive recreational football players over 21 years, to confirm the closure of the epiphyseal plate [21], with MA by roentgenographic evaluation, were included in a cross-sectional study. The control group included recreational football players without MA who visited the clinic for other orthopedic conditions of the foot. Controls were matched to cases according to age and gender.

Foot images were taken using a General Electric Discovery XR656 Plus (General Electric Research, Milwaukee, WI) at a source-to-image distance of 100 cm and were set to 60 kVp and 2.5 mAs. We retrieved the radiographic images using a picture archiving and communication system (PACS) (IMPAX; Agfa Healthcare, Mortsel, Belgium), and radiographic measurements were performed using PACS software and a digital radiographic imaging and measuring system (AutoCad 2019, Autodesk Inc., San Rafael, CA, USA). Dorsoplantar radiographs for weight-bearing conditions were performed independently for each foot with the patients standing with the knee extended. The medial border of the foot was aligned to avoid internal or external rotation of the leg and the foot was pointed straight forward in neutral rotation, parallel to the medial sagittal plane [22]. The X-ray beam was inclined 15° in an anterior-posterior direction centered on the second tarsometatarsal joint at a distance of 100 cm [23,24,25,26,27].

We reviewed 143 weight-bearing dorsoplantar foot radiographs and excluded 63 of them. Inclusion criteria were no evidence of foot trauma that could affect foot anatomy and no history of previous forefoot, midfoot, or rearfoot surgery. Exclusion criteria were being a pregnant woman, having a neurologic disease, lower limb malformation, fracture, or previous surgery of the foot or lower limb, or practicing football less than 5 h/week. We establish two criteria groups: the MA group > 20° [10,28,29] and the control group ≤ 20°. Measurements included 80 radiographs from 40 feet of men (20 in the control group and 20 in the MA group) and 40 feet of women (20 in the control group and 20 in the MA group).

### 2.5. Metatarsus Adductus Measurements

To assess metatarsus adductus, an array of methods has been reported, including: Sgarlato’s method [6,30,31], as shown in Figure 1; the modified Sgarlato’s technique [18,30,31,32], as shown in Figure 2; the Rearfoot angle, calcaneo-second metatarsal angle [30,33], as shown in Figure 3; Root’s angle [5], as shown in Figure 4; Engel’s angle [34], as shown in Figure 5; modified Engel’s [35], as shown in Figure 6; Kite’s angle [4], as shown in Figure 7; Kilmartin’s angle [36], as shown in Figure 8; modified Kirmartin’s angle [14], as shown in Figure 9; Simons’ angle [37], as shown in Figure 10; and Laaveg & Ponseti’s angle [38], as shown in Figure 11.

The following angles were examined on each X-ray: Sgarlato’s angle [6,30,31], Figure 1, is the angle between the longitudinal axis of the 2nd metatarsal and the longitudinal axis of the lesser tarsus using the 4th metatarso-cuboid joint. Line (a) extends between the most medial point of the talo-navicular and the medial cuneiform-first metatarsal joints, while line (b) extends between the most lateral point of the 4th metatarso-cuboid and the calcaneo-cuboid joints. Line (c) extends between the midpoints of lines (a) and (b). Line (d) is perpendicular to line (c) and represents the longitudinal axis of the lesser tarsus. Line (e) represents the longitudinal axis of the second metatarsal bone. Sgarlato’s angle is between lines (d) and (e).

**Figure 1 jcm-11-02043-f001:**
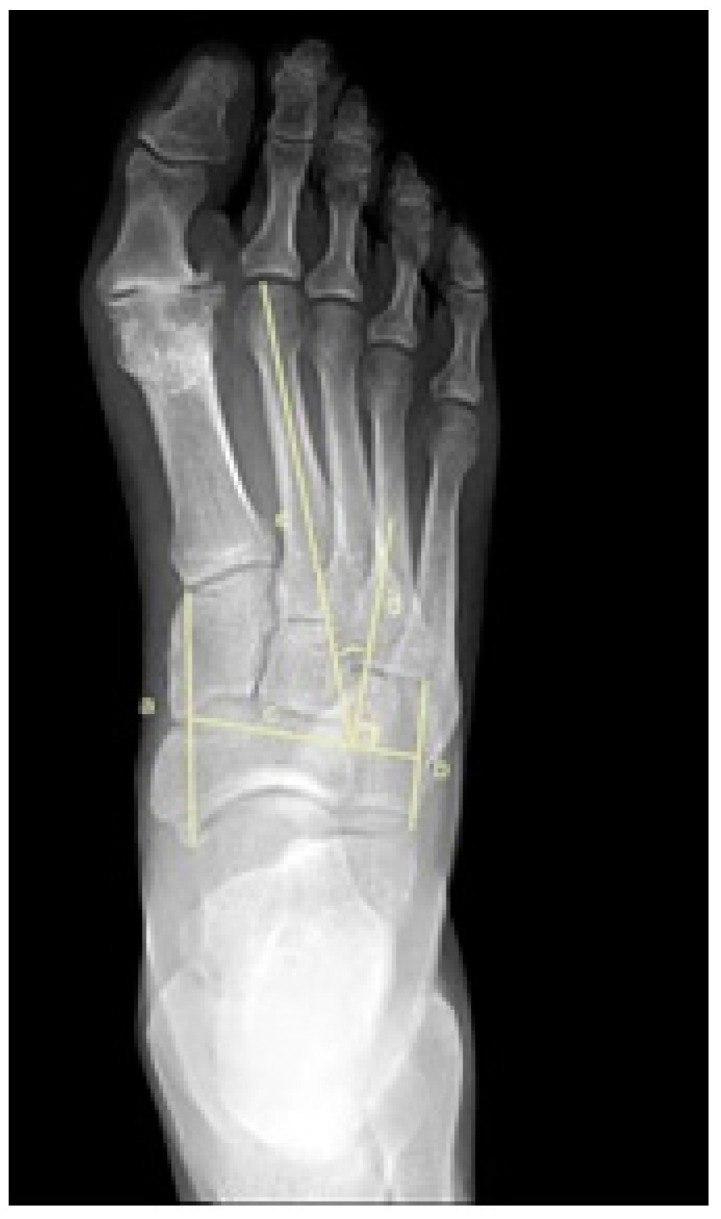
Dorsoplantar weightbearing radiograph of Sgarlato’s angle.

Modified Sgarlato’s [18,30,31,32], Figure 2, is the angle between the longitudinal axis of the second metatarsal and the longitudinal axis of the lesser tarsus using the 5th metatarso-cuboid joint as a reference. Line (a) extends between the most lateral point of the 5th metatarso-cuboid and the calcaneo-cuboid joints. Line (b) extends between the most medial point of the talo-navicular and the medial cuneiform-first metatarsal joints. Line (c) extends between the midpoints of lines (a) and (b). Line (d) represents the longitudinal axis of the second metatarsal bone. Line (e) is perpendicular to line (c) and represents the longitudinal axis of the lesser tarsus. Sgarlato’s angle is between lines (d) and (e).

**Figure 2 jcm-11-02043-f002:**
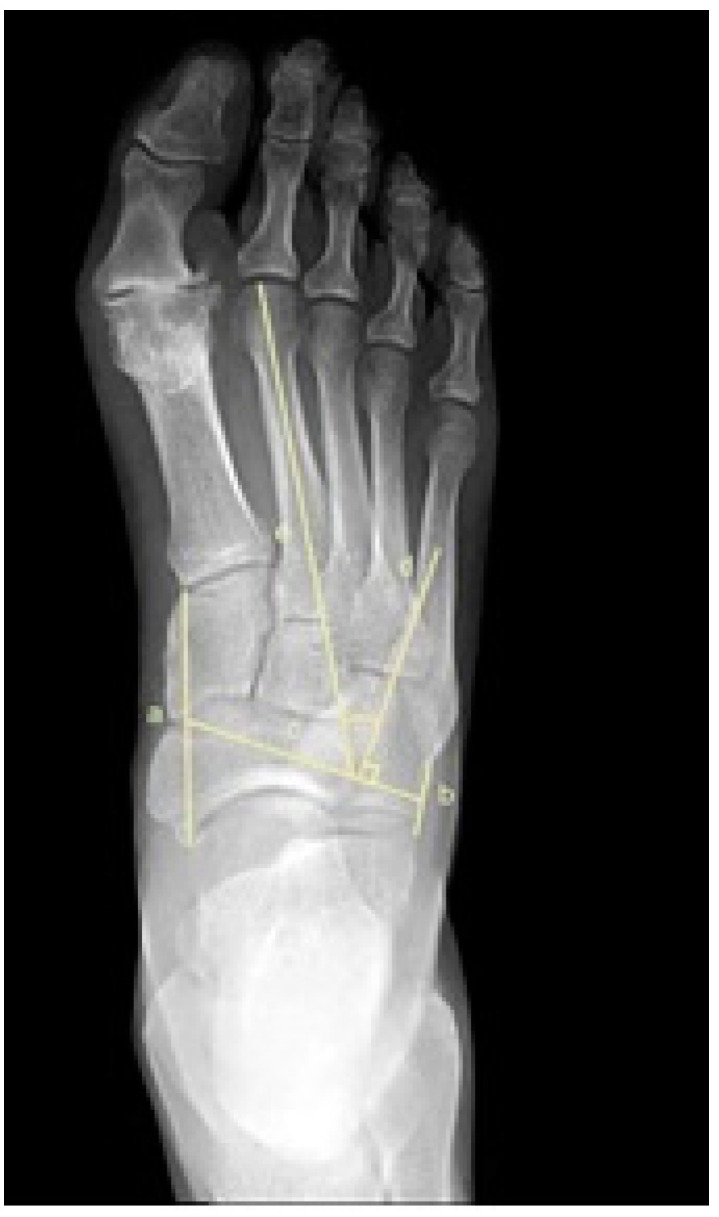
Dorsoplantar weightbearing radiograph of modified Sgarlato’s angle.

The Rearfoot angle [30,33], Figure 3, is the angle between a parallel line to the lateral border of the calcaneum and second metatarsal axis. Line (a) is parallel to the lateral border of the calcaneum, line (b) is parallel to line (a), line (c) axis 2nd metatarsus. The Rearfoot angle is between line (b) and line (c).

**Figure 3 jcm-11-02043-f003:**
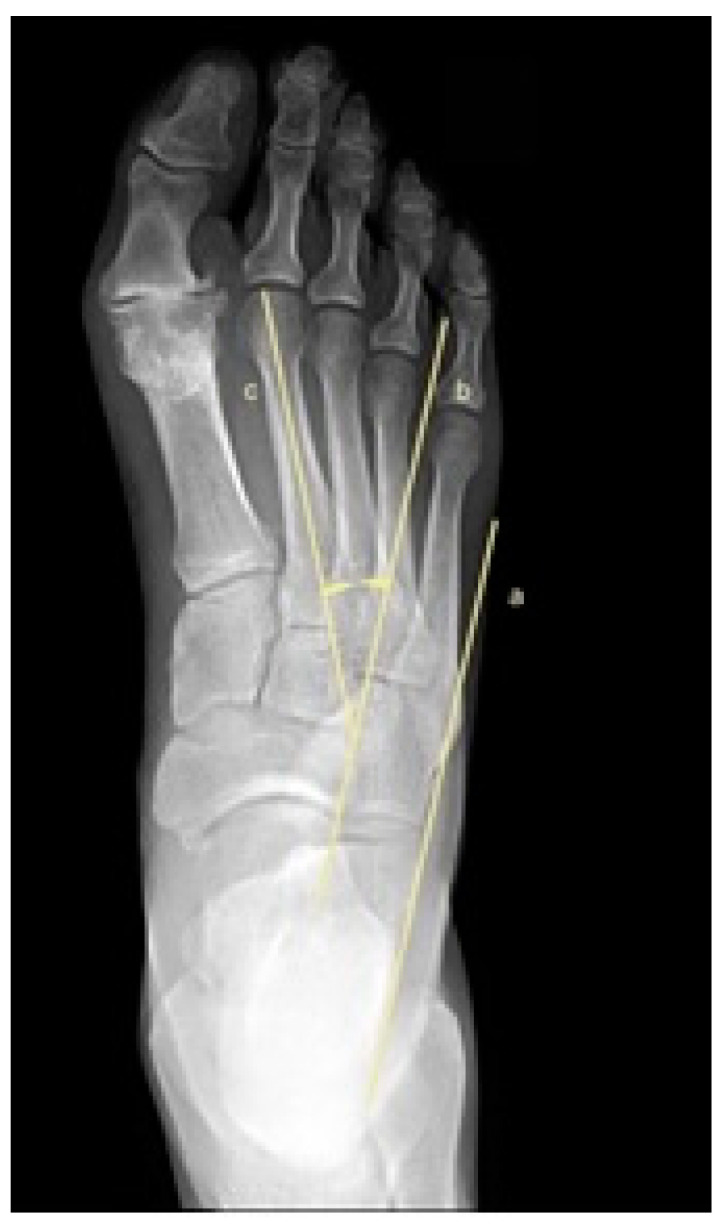
Dorsoplantar weightbearing radiograph of modified Rearfoot angle.

Root’s angle [5], Figure 4, measures the adduction of the forefoot by using an angle formed between the longitudinal axis of the 2nd metatarsus, line (b), and the longitudinal axis of the rearfoot, line (a).

**Figure 4 jcm-11-02043-f004:**
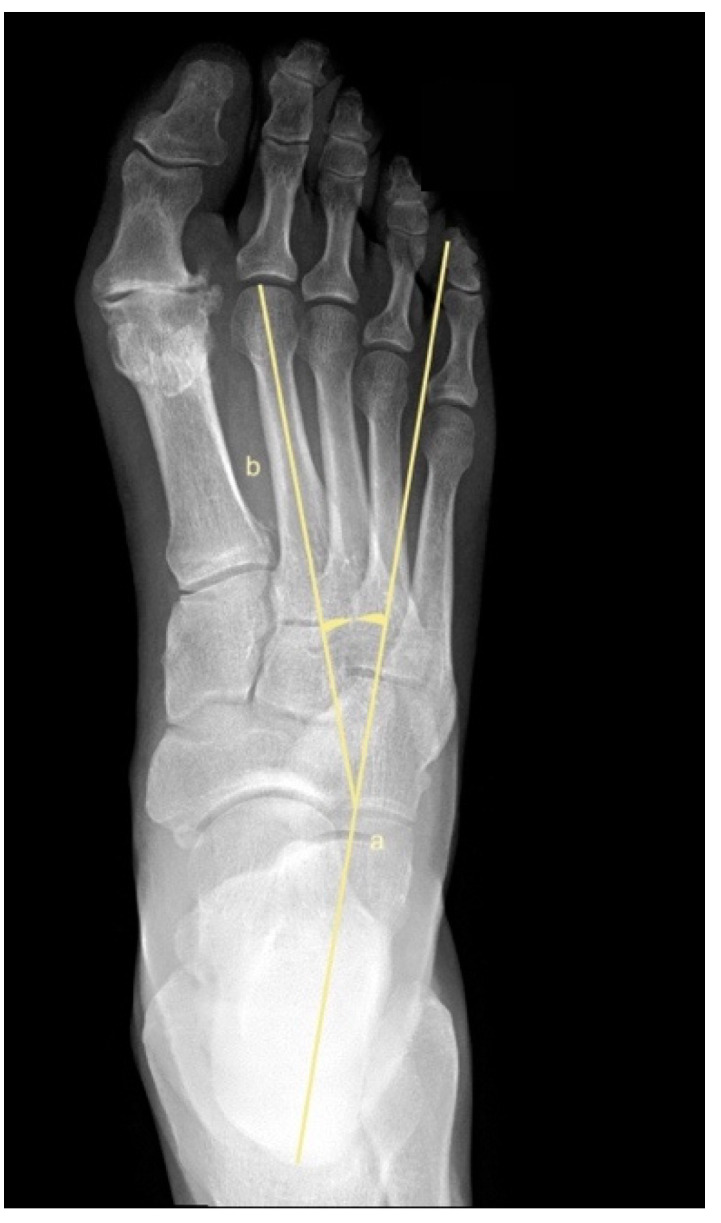
Dorsoplantar weightbearing radiograph of Root’s angle.

Engel’s angle [34], Figure 5, is the angle between the longitudinal axis of the second cuneiform as shown at line (b) and the longitudinal axis of the 2nd metatarsal bone as shown at line (a).

**Figure 5 jcm-11-02043-f005:**
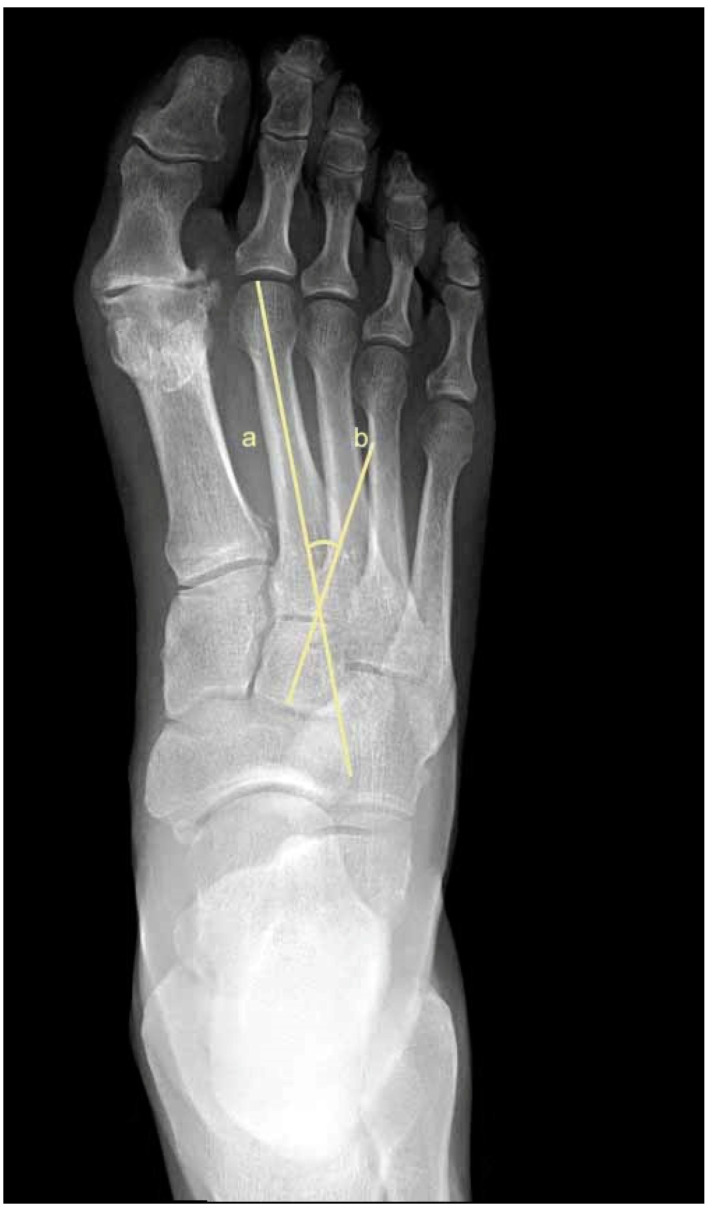
Dorsoplantar weightbearing radiograph of Engel’s angle.

Modified Engel’s angle [35], Figure 6, is the angle between the longitudinal axis of the 2nd metatarsal, line (c), and a line perpendicular to the proximal articular surface of the cuneiform II, line (b) parallel line of proximal side cuneiform II, line (a). Modified Engel’s angle is the angle between line (b) and line (c).

**Figure 6 jcm-11-02043-f006:**
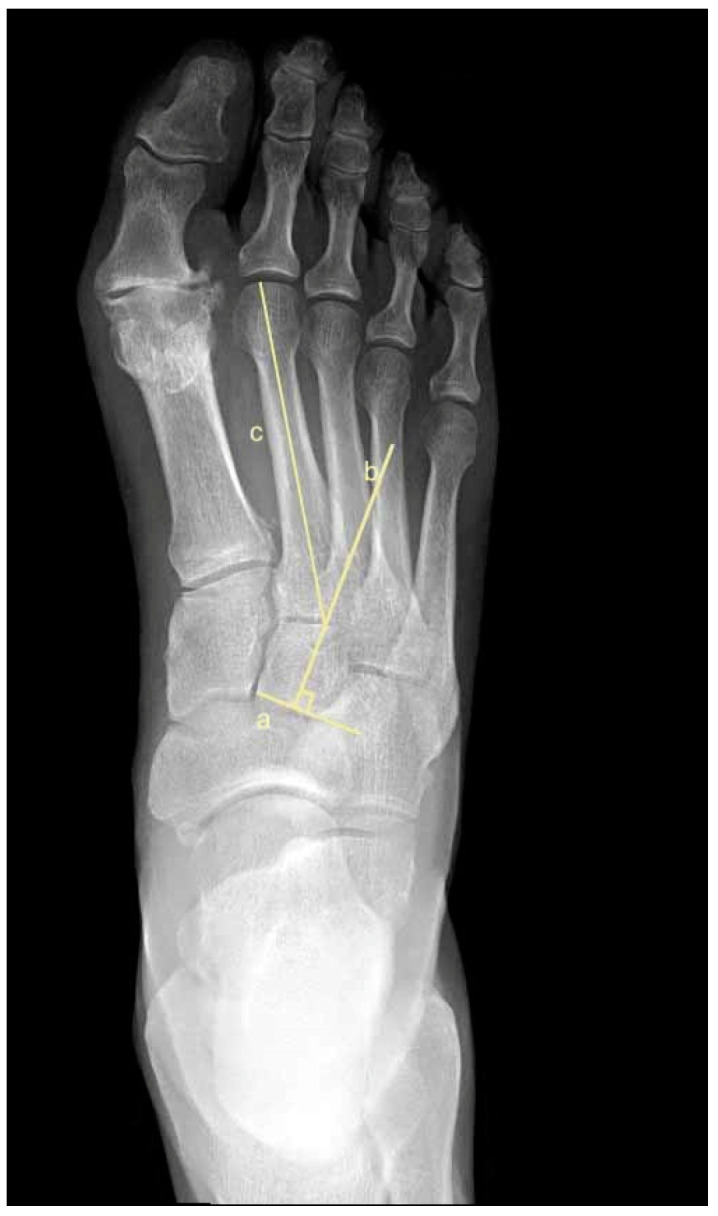
Dorsoplantar weightbearing radiograph of modified Engel’s angle.

Kite’s angle [4], Figure 7, is the angle between a line perpendicular of the articular surface of the talar head (longitudinal axis of the talus) and a parallel line of the lateral border of the calcaneum. Line (a) is parallel to the lateral border of the calcaneum, line (b) is parallel to line (a), and line (c) connects the extremes of the articular surface of the talar head. Line (d) is perpendicular to line (c) and represents the longitudinal axis of the talus. Kite’s angle is the angle between line (b) and line (d).

**Figure 7 jcm-11-02043-f007:**
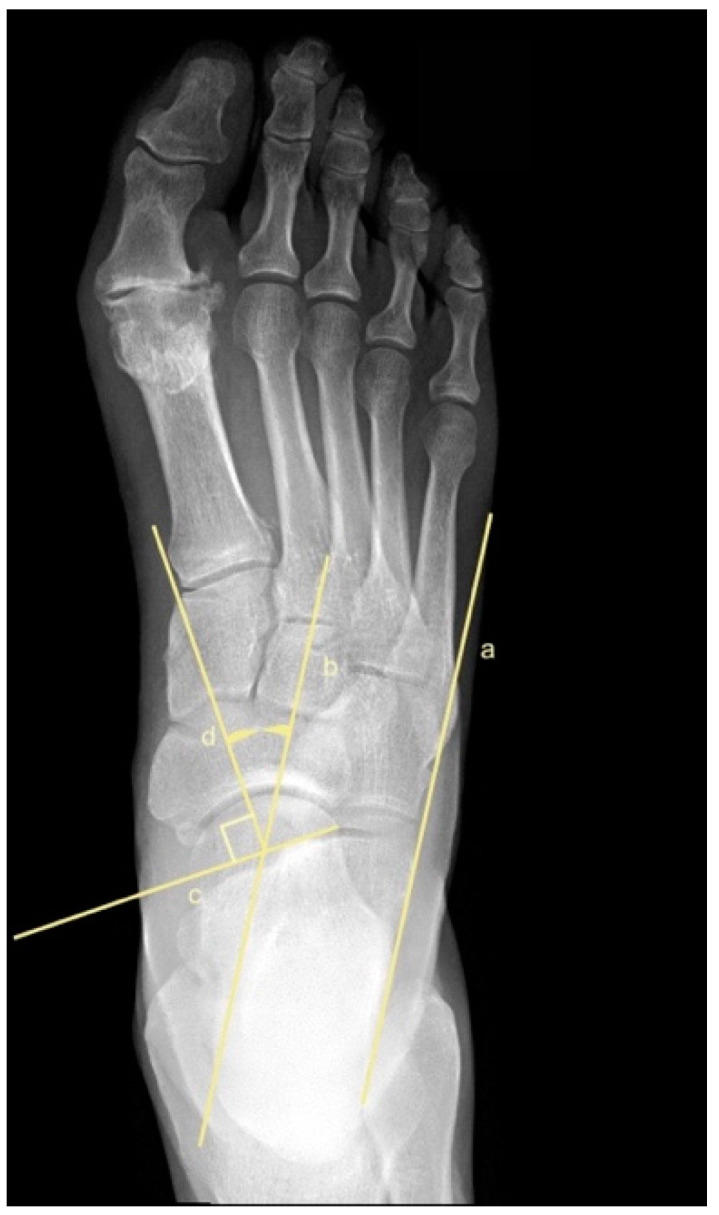
Dorsoplantar weightbearing radiograph of Kite’s angle.

Kilmartin’s angle [36], Figure 8, is measured between a line parallel to the lateral border of the calcaneum and the 1st metatarsus axis. Line (a) is parallel to the lateral border of the calcaneum. Line (b) is a line parallel to line (a); line (c) is the axis of the 1st metatarsal. Kilmartin’s angle is the angle between line (b) and line (c).

**Figure 8 jcm-11-02043-f008:**
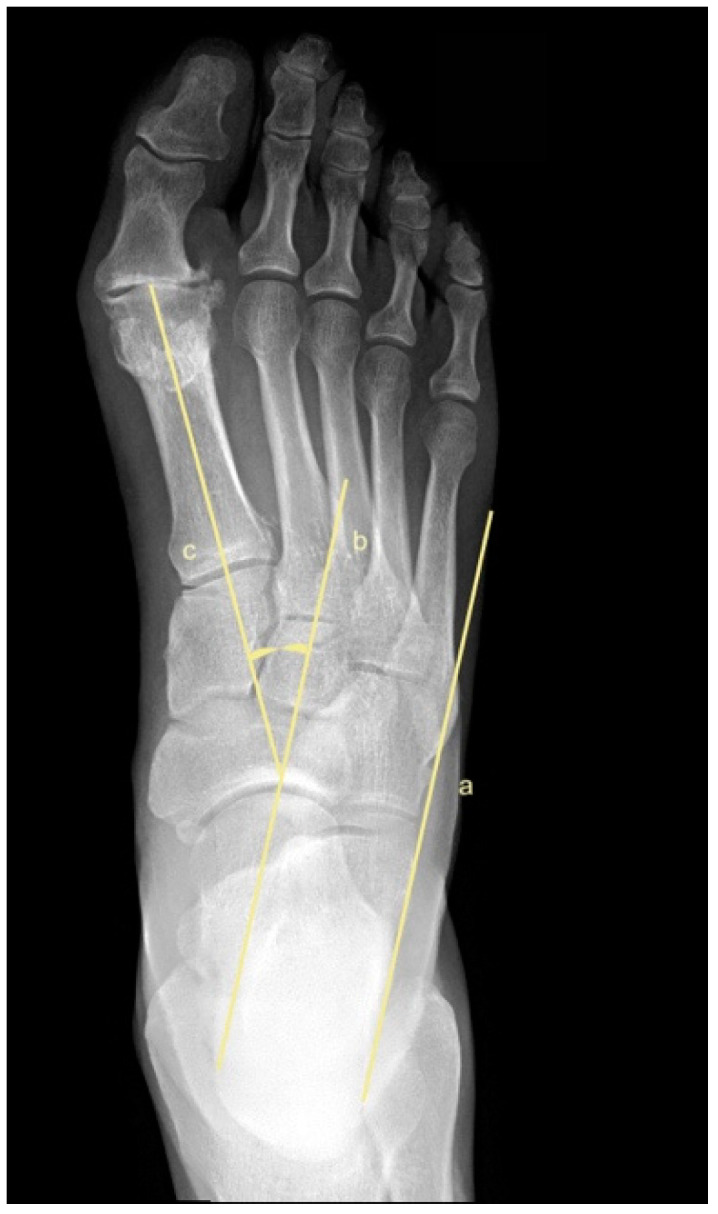
Dorsoplantar weightbearing radiograph of Kilmartin’s angle.

Modified Kilmartin’s angle [14], Figure 9, is the angle measured between the longitudinal axis of the second metatarsal and the transverse axis of the lesser tarsus. Line (a) extends between the most medial point of the talo-navicular and the medial cuneiform-first metatarsal joints. Line (b) extends between the most lateral point of the 4th metatarso-cuboid and the calcaneo-cuboid joints. Line (c) extends between the midpoints of lines (a) and (b). Line (d) represents the longitudinal axis of the second metatarsal bone. Modified Kilmartin’s angle is the angle between lines (c) and (d).

**Figure 9 jcm-11-02043-f009:**
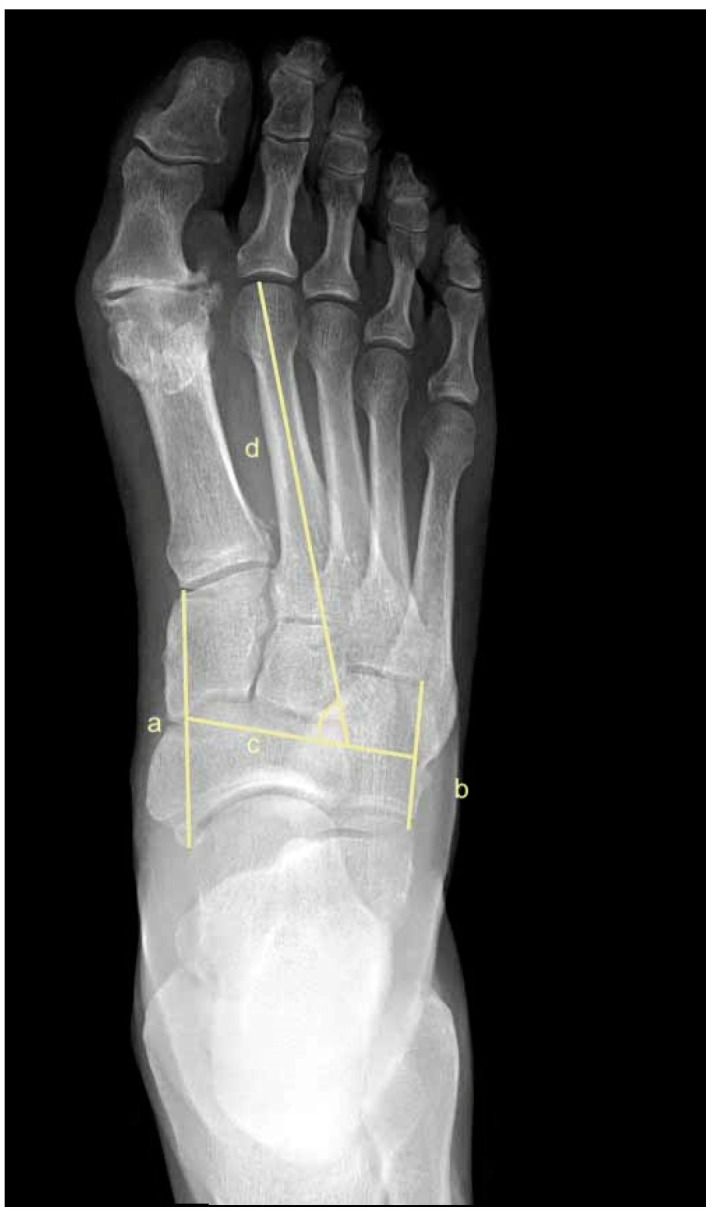
Dorsoplantar weightbearing radiograph of modified Kilmartin’s angle.

Simons’ angle [37], Figure 10, is the angle between a perpendicular line (b) of the articular surface of the talar head (longitudinal axis of the talus) line (a) and the longitudinal axis of the 1st metatarsal, line (c).

**Figure 10 jcm-11-02043-f010:**
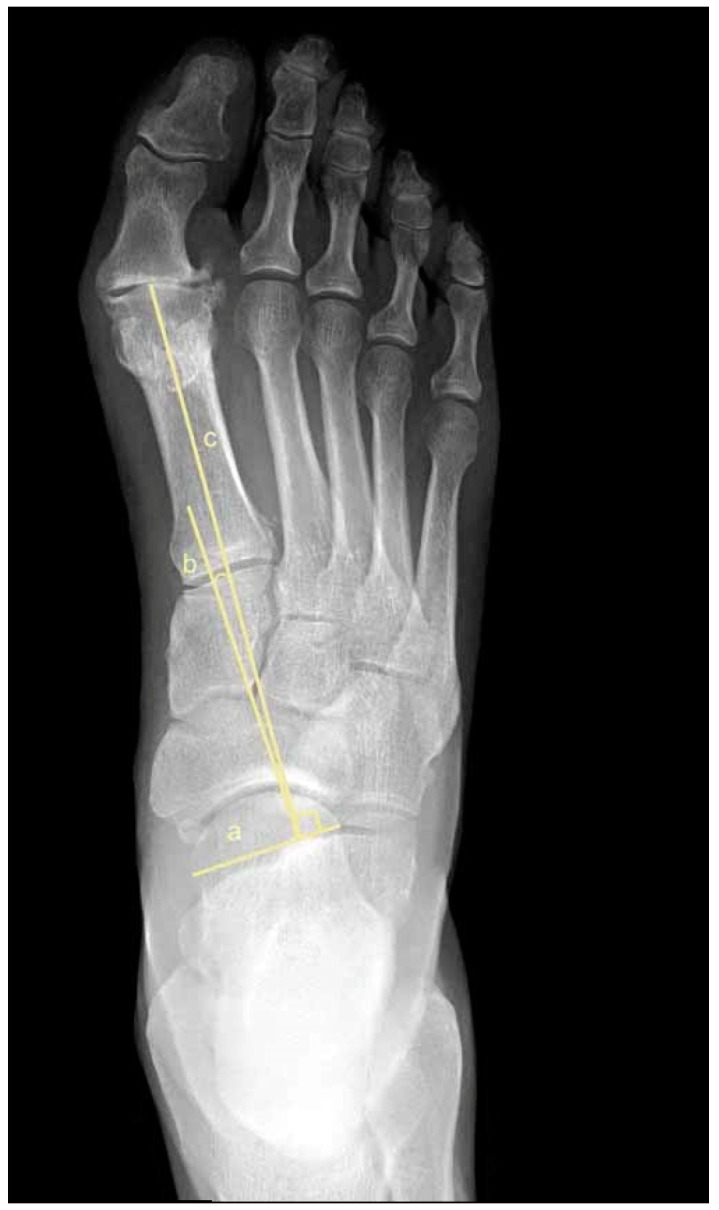
Dorsoplantar weight-bearing radiograph of Simons’ angle.

Laaveg & Ponseti’s angle [38], Figure 11, is the angle between a line parallel to the lateral surface of calcaneum bone, line (a), and the longitudinal axis of the 5th metatarsal bone, line (b).

**Figure 11 jcm-11-02043-f011:**
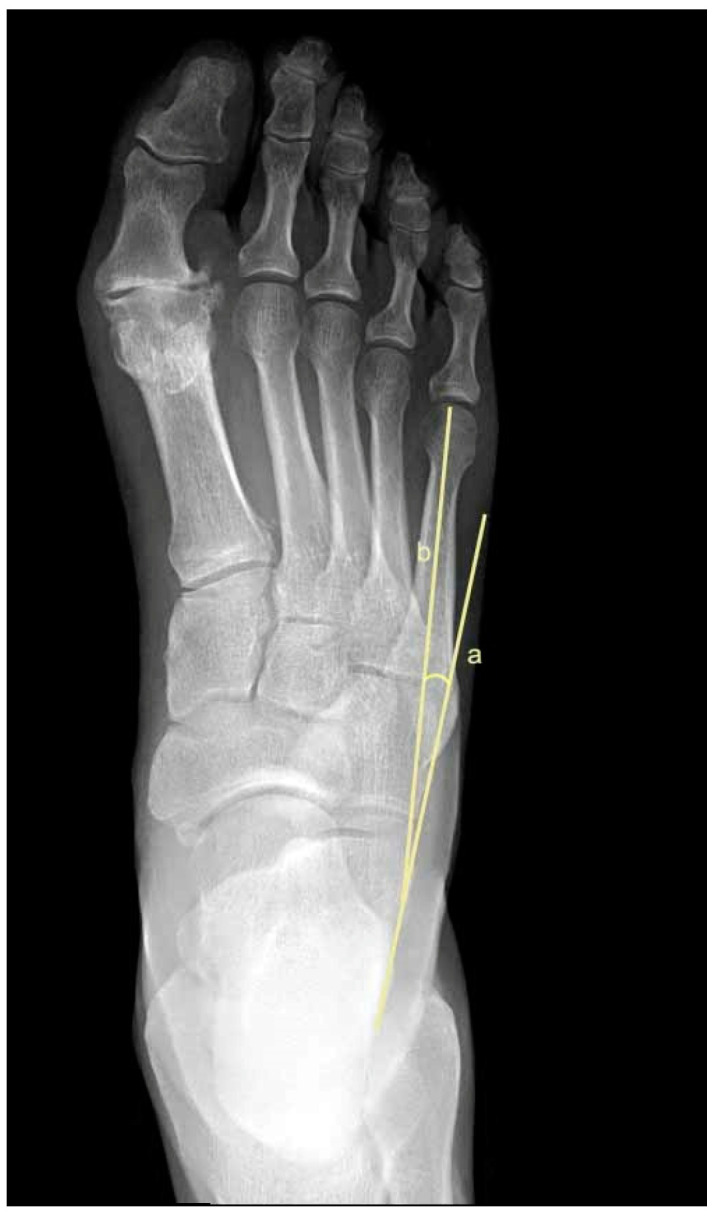
Dorsoplantar weight-bearing radiograph of Laaveg & Ponseti’s angle.

#### 2.5.1. Reliability Study Protocol

Intra and inter-rater reliability study: All radiographs were selected for inclusion in the intra-rater reliability study. Two evaluators conducted all measurements for each radiograph at three different times, separated by 1 week each. In total, 6 measurements in each of the different angles of the Rx were taken.

Before each measurement session, the order of the patients was randomized so that the radiographs viewed would not be in the same order as that of the previous week’s measurements.

To guarantee the quality of the data and provide a benchmark for the reproducibility of each measurement, the two readers independently measured each angle on a radiograph in two different cities. The readers were physicians with more than 15 years of radiology experience and were trained on an initial set of three cases from the sample under the direction of another senior physician with more than 30 years of radiology experience. All measurements were recorded on a Microsoft Excel spreadsheet.

#### 2.5.2. Statistical Analysis

Quantitative data were described as mean ± standard deviation (SD) as well as lower and upper limits for a 95% CI, and median and interquartile range

All variables were examined for normality of distribution using the Kolmogorov-Smirnov test; data were considered normally distributed if *p* > 0.05. Independent Student *t*-tests were performed to determine whether differences were statistically significant when a normal distribution was shown. Measurements that were not normally distributed were tested using a non-parametric U Mann Whitney test.

Reliability between two measurement values was determined using the Intraclass Correlation Coefficient two-way random effects, absolute agreement, multiple raters/measurements (ICC 2,k), and Pearson´s correlation coefficient (r). The average of two trials for each test session on each radiograph was used to calculate intersession reliability. ICC values were interpreted as poor (ICC < 0.40), fair (ICC = 0.40–0.59), good (ICC = 0.60–0.74), and excellent (ICC = 0.75–1.0) [39]. According to the recommendations of Portney and Watkins, clinical measurements with reliability coefficients greater than 0.90 improve the probability that the measure is valid [40].

Coefficients of variation (CVs) were analyzed for the absolute comparison of parameters. The CVs were calculated to test the intra-session reliability. The CV was calculated as the mean normalized to the SD. A higher CV value shows that the data are more heterogeneous.

In addition, r coefficient values were categorized as weak (r = 0.00–0.40), moderate (r = 0.41–0.69), and strong (r = 0.70–1.00) [41].

The 95% limits of agreement (LoA) between sessions and devices expressed the degree of error proportional to the mean of the measurement units, and these statistics were calculated using the methods described by Bland and Altman [42]. If the differences between the measurements tended to agree, the results were close to zero.

Standard errors of measurement (SEM) were calculated to measure the range of error of each parameter. SEM were calculated from the ICCs and SDs for each of the three measurements, according to the formula SEM = SD × sqrt (1–ICC).

The minimum detectable change (MDC) at a confidence level of 95%, which reflects the magnitude of change necessary to provide confidence that a change is not the result of random variation or measurement error, was calculated from the SEM values by the following formula: MDC = $\sqrt{2}\times 1.96\times SEM$. Both SEM and MDC were analyzed according to Bland and Altman [42].

In all of the analyses, statistical significance was established by a *p* value of less than 0.05, with an interval of confidence of 95%, and analyses were performed with commercially available software (SPSS 25.0, SPSS Inc., Chicago, IL, USA).

## 3. Results

In total, 80 participants were recruited. Descriptive data of the age variable showed a normal distribution (*p* = 0.216), as shown in Table 1.

Table 2 shows the analysis of the reliability of the metatarsus adductus angle by the first observer. Excellent values were obtained on the first and second sessions with ICC > 0.910 for all variables. Reliability intersession values ranged from [ICC = 0.913(0.864–0.944)] to [ICC = 0.985(0.977–0.990)]. A strong correlation was found in all angles between the first and second sessions, ranging from r = 0.847 to r = 0.988.

SEM and CV values were very low for all measurements, as well as MCD, except for Kite’s angle, with MCD = 5.698. We found systematic differences between sessions on Engel’s angle, modified Engel’s angle, modified Kilmartin’s angle, and Laaveg & Ponseti’s angle (*p* < 0.05).

Table 3 shows the analysis of reliability of the metatarsus adductus angle by the second observer. Excellent values were obtained in the first and second sessions with ICC > 0.900 for the most variables except Kite’s angle [ICC = 0.413 (0.149–0.605)], modified Engel’s angle [ICC = 0.872 (0.814–0.914)], and modified Kilmartin’s angle [ICC = 0.887 (0.837–0.924)], Reliability intersession values ranged from [ICC = 0.961 (0.940–0.975)] to [ICC = 0.997 (0.996–0.998)], except Kite’s angle with [ICC = 0.751 (0.611–0.840)], A strong correlation was found in all angles between the first and second sessions ranging from r = 0.926 to r = 0.995.

SEM was low for all variables except Kite’s angle, SEM = 7.876, MCD on modified Engel’s angle was 7.199, while that for the rest of variables was low. CV was low for all variables. We found systematic differences between sessions on Rearfoot’s angle and Kilmartin’s angle (*p* < 0.05).

Table 4 shows an analysis of the reliability of metatarsus adductus angle measurements between observers. Excellent ICC values were obtained between observers ICC > 0.900 for all variables except Kite’s angle and modified Root’s angle, [ICC = 0.854 (0.706–0.919)] and [ICC = 0.860 (0.728–0.921)], respectively. A strong correlation was found in all angles between the first and second observers, ranging from r = 0.791 and r = 0.952.

Low SEM was found for all variables. CV was low for all variables except modified Kilmartin’s angle (CV = 11.442). MCD was low for all variables except Root’s angle (MCD = 7.312), Kite’s angle (MCD = 7.541), and Simon’s angle (MCD = 7.084). The LoA were low for all variables except Root’s angle and Kite’s angle [LoA = 2.4 (−7.1–11.8)] and [LoA = 2.5 (−7.1–12.2)], respectively. We found systematic differences inter-observer on kite’s angle and modified Kilmartin’s angle (*p* < 0.05).

## 4. Discussion

Lack of consensus among physicians could be due to the lack of agreement regarding classification. Metatarsus adductus measurement should be simple, easy to remember, and effective for all physicians. Various radiographic measurements have been developed and used for evaluating metatarsus adductus [5,14,30,32,33,43]. The objective of this study was to update guidelines regarding the inter-observer reliability of the common measurement most currently used in MA assessment [30,32,34,35]. Before discussing the findings of our study, we must indicate limitations. Metatarsus adductus is a complex three-dimensional deformity, and we only can explain with this study in 2-dimensional plane.

The intra and inter-observer reliability of common angular measurements of various foot disorders has been reported to be satisfactory [14]. The reliability of the angular measurements used to evaluate MA has been reported to a much lesser extent than other radiological foot measurements [44].

In each session, the first and second observers, and also the inter-observer, present high ICC the following measurements: Sgarlato and modified Sgarlato, both with ICC > 0.900; in addition, we found low SEM and CV. No systematic differences were found between sessions on these measurements. Sgarlato and modified Sgarlato’s angles’ ICC values are in concordance with the values shown by Dominguez [18] and Dawoodi [44].

On the other hand, the measurement angles for Root, modified Engel, Kite, and modified Kilmartin show an ICC < 0.900 in any session, intersession, or inter-observer and should be higher, as indicated by Portney [40].

Likewise, other angle measurements, like Engel, modified Engel, Rearfoot, modified Kilmartin, Kite, and Root, showed systematic differences in any intersession or inter-observer measurements. Simon’s angle presents a high MCD value between observers

### Limitation of This Study

This study has been achieved in adults; thus, further research is needed to find intra and inter-observer MA in children. Furthermore, computed tomography (CT) scan is a useful diagnostic tool to produce a 3D image of bones; further research is needed to find the reliability and repeatability of metatarsus adductus angles as a new method, as postulated by Spinarelli A et al. [45].

## 5. Conclusions

Our study suggests that different techniques used in assessing metatarsus adductus demonstrated high values of intra and inter-observer reliability, Sgarlato and modified Sgarlato’s angle like in other studies. On the other hand, we do not advise using the measures Root, Rearfoot, Engel, modified Engel, Kilmartin, modified Kilmartin, Laaveg & Ponseti, and Kite for presenting an ICC lower than 0.900 or systematic differences between intersession and/or inter-observers, and Simons for presenting a high MCD value.

## Figures and Tables

**Table 1 jcm-11-02043-t001:** Age of the participants by sex distribution.

Variables	Total (N = 80)		Male (n = 40)			Female (n = 40)			
Descriptive Data	Mean ± SD (95% CI)	Median (IR)	Mean ± SD (95% CI)	Median (IR)	*p* K-S	Mean ± SD (95% CI)	Median (IR)	*p* K-S	*p* Value
Age	29.48 ± 5.12 (28.34–30.62)	29.00 (7.50)	30.20 ± 6.00 (28.27–32.12)	29.00 (7.75)	0.018	28.77 ± 4.01 (27.49–30.05)	28.00 (6.75)	0.200	0.216

Abbreviations: MA, Metatarsus Adductus; N, sample size; SD, standard deviation; CI, confidence interval; IR, Interquartile range; *p* K-S, Kolgomorov-Smirnov test and *p* > 0.05 considered normal distribution; *p* value, from U Mann Whitney for independent group. Statistical significance for a *p* value < 0.05, with a 95% confidence interval.

**Table 2 jcm-11-02043-t002:** Analysis of metatarsus adductus angle measurements between the first and second sessions by the first observer and normalized values.

FIRST OBSERVER MA MEASUREMENTS																				
SESSIONS	FIRST SESSION							SECOND SESSION							INTERSESSION					
Angle	Mean ± SD (CI95%)	ICC (CI95%)	Median (IR)	SEM	CV	MCD	*p* K-S	Mean ± SD (CI95%)	ICC (CI95%)	Median (IR)	SEM	CV	MCD	*p* K-S	Mean ± (SD) (CI95%)	ICC (CI95%)	Median (IR)	SEM	R (p)	*p*-Value
Sgarlato	18.14 ± 5.83 (16.84–19.44)	0.988 (0.982–0.992)	19.62 (9.48)	0.395	0.487	1.094	0.001	17.90 ± 5.82 (16.60–19.20)	0.977 (0.967–0.985)	19.32 (7.89)	0.869	0.325	2.409	0.001	18.02 ± 5.76 (16.74–19.30)	0.970 (0.962–0.984)	18.92 (8.56)	0.911	0.924 ** 0.001	0.182 **
Modified Sgarlato	24.07 ± 6.03 (22.73–25.42)	0.986 (0.976–0.991)	25.33 (9.29)	0.690	0.242	1.912	0.059	24.2 ± 6.15 (22.83–25.57)	0.988 (0.982–0.992)	24.5 (9.93)	0.674	0.254	1.867	0.200	24.13 ± 6.02 (22.79–25.48)	0.979 (0.967–0.986)	24.94 (10.07)	0.872	0.958 * 0.001	0.539 *
Rearfoot	14.94 ± 8.80 (12.98–16.90)	0.977 (0.966–0.984)	16.16 (14.02)	1.335	0.589	3.699	0.077	15.60 ± 8.66 (13.67–17.52)	0.985 (0.979–0.990)	17.37 (12.9)8	1.061	0.555	2.940	0.176	15.27 ± 8.49 (13.38–17.16)	0.943 (0.911–0.963)	16.91 (13.66)	2.027	0.893 * 0.001	0.149 *
Root	16.19 ± 6.62 (14.72–17.66)	0.945 (0.920–0.963)	17.8 (10.01)	1.553	0.409	4.303	0.003	15.50 ± 6.45 (14.06–16.94)	0.967 (0.952–0.978)	16.4 (8.41)	1.172	0.416	3.248	0.040	15.84 ± 6.31 (14.44–17.25)	0.927 (0.886–0.953)	16.60 (9.64)	1.705	0.851 ** 0.001	0.126 **
Engel	25.45 ± 6.01 (24.11–26.78)	0.975 (0.964–0.983)	25.33 (6.84)	0.950	0.236	2.634	0.200	26.22 ± 6.22 (24.84–27.61)	0.989 (0.984–0.993)	26.67 (9.36)	0.652	0.237	1.808	0.200	25.83 ± 5.94 (24.51–27.16)	0.935 (0.897–0.959)	26.13 (8.06)	1.514	0.885 * 0.001	0.021 *
Modified Engel	25.47 ± 6.98 (23.92–27.03)	0.979 (0.968–0.986)	25.33 (8.96)	1.011	0.274	2.803	0.200	24.68 ± 7.21 (23.08–26.29)	0.991 (0.987–0.994)	24.53 (10.67)	0.684	0.292	1.896	0.086	25.08 ± (6.95) (23.53–26.62)	0.953 (0.925–0.970)	25.02 (10.17)	1.290	0.915 * 0.001	0.018 *
Kite	24.98 ± 6.93 (23.43–26.52)	0.912 (0.872–0.940)	24.25 (10.09)	2.056	0.277	5.698	0.200	25.54 ± 7.86 (23.79–27.29)	0.961 (0.943–0.974)	24.8 (11.7)	1.552	0.308	4.303	0.056	25.26 ± 7.11 (23.67–26.84)	0.913 (0.864–0.944)	24.61 (10.40)	2.097	0.847 * 0.001	0.236 *
Kilmartin	22.93 ± 9.93 (20.72–25.14)	0.985 (0.979–0.990)	24.31 (3.29)	1.216	0.433	3.371	0.042	23.47 ± 10.12 (21.22–25.72)	0.992 (0.988–0.995)	25.48 (11.66)	0.905	0.431	2.509	0.020	23.20 ± 9.88 (21.00–25.40)	0.970 (0.953–0.981)	25.46 (12.36)	1.711	0.917 ** 0.001	0.315 **
Modified Kilmartin	69.49 ± 5.85 (68.19–70.79)	0.980 (0.971–0.987)	69.91 (8.01)	0.827	0.084	2.293	0.200	69.68 ± 6.03 (68.64–71.32)	0.992 (0.988–0.994)	69.91 (8.68)	0.539	0.087	1.495	0.200	69.74 ± 5.88 (68.42–71.05)	0.979 (0.966–0.987)	70.00 (7.43)	0.852	0.963 * 0.001	0.008 *
Simons	2.78 ± 9.33 (0.70–4.85)	0.996 (0.950–0.977)	3.09 (13.05)	1.720	3.356	4.769	0.200	2.57 ± 9.49 (0.46–4.68)	0.981 (0.973–0.987)	2.96 (13.54)	1.308	3.693	3.626	0.200	2.67 ± 9.34 (0.59–4.75)	0.985 (0.977–0.990)	2.98 (13.14)	1.144	0.988 * 0.001	0.403 *
Laaveg & Ponseti	−4.54 ± 7.70 (−6.25–−2.82)	0.991 (0.988–0.994)	−3.82 (10.57)	0.730	−1.696	2.025	0.200	−3.91 ± 7.99 (−5.69–−2.13)	0.965 (0.949–0.976)	−3.18 (11.22)	1.495	−2.043	4.143	0.200	−4.22 ± 7.74 (−5.95–−2.50)	0.972 (0.956–0.982)	−4.31 (11.05)	1.295	0.951 * 0.001	0.019 *

Abbreviations: MA, Metatarsus Adductus; CI Confidence Interval; SD, standard deviation; IR, Interquartile Range; SEM, Standard Error of Measurement; MCD, Minimum Detectable Change; CV, Coefficient of Variation; ICC, Intraclass Correlation Coefficient; r, ** Spearman and * Pearson correlation coefficient; SEM, standard error of measurement; *p* K-S, Kolgomorov Smirnov test and *p* > 0.05 considered normal distribution; * *p* value from Paired *t*-test, ** *p* value from Wilcoxon signed rank test; Statistical significance for a *p* value < 0.05, with a 95% confidence interval.

**Table 3 jcm-11-02043-t003:** Analysis of metatarsus adductus angle measurements between the first and second sessions by the second observer and normalized values.

SECOND OBSERVER MA MEASUREMENTS																				
SESSIONS	FIRST SESSION							SECOND SESSION							INTERSESSION					
Angles	Mean ± SD (CI 95%)	ICC (CI 95%)	Median (IR)	SEM	CV	MCD	*p* K-S	Mean ± SD (CI95%)	ICC (CI95%)	Median (IR)	SEM	CV	MCD	*p* K-S	Mean ± SD (CI95%)	ICC (CI 95%)	Median (IR)	SEM	r (*p*)	*p* Value
Sgarlato	18.49 ± 6.33 (17.08–19.90)	0.998 (0.997–0.999)	19.95 (9.12)	0.283	0.342	0.784	0.019	18.54 ± 5.95 (17.22–19.87)	0.996 (0.995–0.998)	19.88 (9.16)	0.376	0.320	1.043	0.003	18.52 ± 6.10 (17.16–19.87)	0.985 (0.977–0.990)	19.89 (9.17)	0.747	0.982 ** (0.001)	0.735 **
Modified Sgarlato	24.03 ± 6.31 (22.63–25.44)	0.967 (0.952–0.978)	24.77 (8.02)	1.146	0.262	3.177	0.200	24.12 ± 6.16 (22.75–25.50)	0.996 (0.994–0.997)	24.63 (7.43)	0.390	0.255	1.079	0.200	24.08 ± 6.20 (22.70–25.46)	0.990 (0.984–0.994)	24.83 (7.61)	0.620	0.981 * (0.001)	0.516 *
Rearfoot	15.96 ± 8.62 (14.04–17.88)	0.991 (0.986–0.994)	18.02 (13.08)	0.818	0.540	2.226	0.001	15.51± 8.79 (13.55–17.46)	0.996 (0.994–0.997)	17.34 (13.45)	0.556	0.566	1.541	0.006	15.73 ± 8.62 (13.81–17.65)	0.981 (9.970–9.988)	17.65 (12.89)	1.188	0.969 ** (0.001)	0.007 **
Root	18.28 ± 8.34 (16.42–20.14)	0.981 (0.973–0.988)	19.29 (12.46)	1.150	0.456	3.186	0.200	18.15 ± 8.60 (16.23–20.06)	0.996 (0.995–0.998)	18.8 (11.67)	0.544	0.473	1.507	0.200	18.21 ± 8.44 (16.33–20.09)	0.981 (9.970–0.988)	18.21 (12.09)	0.706	0.986 * (0.001)	0.410 *
Engel	24.13 ± 6.43) (22.70–25.56)	0.995 (0.993–0.997)	24.47 (6.92)	0.455	0.266	1.260	0.200	24.09 ± 6.30 (22.69–25.49)	0.997 (0.996–0.998)	24.44 (7.29)	0.345	0.261	0.956	0.200	24.11 ± 6.36 (22.70–25.53)	0.997 (0.996–0.998)	24.49 (6.99)	0.348	0.995 * (0.001)	0.575 *
Modified Engel	23.57 ± 7.26) (21.95–25.18)	0.872 (0.814–0.914)	23.56 (9.47)	2.597	0.308	7.199	0.200	23.52 ± 7.16 (21.93–25.12)	0.998 (0.997–0.999)	23.51 (9.38)	0.320	0.304	0.887	0.200	23.54 ± 7.08 (21.97–25.12)	0.961 (0.940–0.975)	23.58 (9.39)	1.398	0.926 * (0.001)	0.889 *
Kite	28.25 ± 10.28 (25.96–30.54)	0.413 (0.149–0.605)	27.39 (10.21)	7.876	0.363	2.183	0.014	27.36 ± 7.18 (25.76–28.96)	0.996 (0.994–0.997)	27.28 (9.68)	0.454	0.262	1.258	0.200	27.81 ± 7.93 (26.04–29.57)	0.751 (0.611–0.840)	27.81 (19.19)	3.957	0.951 ** (0.001)	0.341 **
Kilmartin	22.37 ± 9.56 (20.24–24.50)	0.982 (0.973–0.988)	24.51 (12.69)	1.283	0.427	3.555	0.004	22.46 ± 9.38 (20.37–24.55)	0.998 (0.997–0.999)	24.9 (12.42)	0.419	0.417	1.162	0.001	22.42 ± 9.45 (20.31–24.52)	0.995 (0.992–0.997)	24.66 (12.93)	0.668	0.990 ** (0.001)	0.038 **
Modified Kilmartin	67.76 ± 6.31 (66.35–69.19)	0.993 (0.990–0.995)	68.51 (9.02)	0.528	0.093	1.463	0.200	67.38 ± 6.61 (65.91–68.85)	0.887 (0.837–0.924)	68.19 (8.31)	2.222	0.098	6.159	0.200	67.57 ± 6.36 (66.15–68.99)	0.968 (0.950–0.979)	68.43 (8.74)	1.138	0.939 * (0.001)	0.141 *
Simons	2.30 ± 9.85 (0.10–4.49)	0.998 (0.997–0.999)	3.63 (14.05)	0.441	4.283	1.221	0.200	2.38 ± 9.95 (0.17–4.60)	0.998 (0.998–0.999)	3.79 (14.73)	0.445	4.181	1.233	0.091	2.34 ± 9.89 (0.14–4.54)	0.997 (0.996–0.998)	3.71 (14.30)	0.542	0.995 * (0.001)	0.885 *
Laaveg & Ponseti	−4.58 ± 8.05 (−6.38–−2.79)	0.975 (0.964–0.983)	−4.36 (9.70)	1.273	−1.758	3.528	0.200	−4.85 ± 9.14 (−6.89–−2.82)	0.963 (0.946–0.975)	−4.38 (9.60)	1.847	−4.380	5.119	0.200	−4.72 ± 8.30 (−6.57–−2.87)	0.924 (0.882–0.951)	−4.47 (9.88)	2.288	0.865 * (0.001)	0.100 *

Abbreviations: SD, standard deviation; CI, Confidence Interval; ICC, Intraclass Correlation Coefficient; SEM, Standard Error of Measurement; MCD, Minimum Detectable Change; LoA, 95% limits of agreement; *p* K-S, Kolgomorov Smirnov test and *p* > 0.05 considered normal distribution; r, ** Spearman and * Pearson correlation coefficient; * *p* value from U Mann Whitney; ** *p* value from Independent *t* student test; Statistical significance for a *p* value < 0.05, with a 95% confidence interval.

**Table 4 jcm-11-02043-t004:** Analysis of the reliability of metatarsus adductus angle measurements between observers.

Observer	First Observer Intersession		Second Observer Intersession		Interobserver							
Variables	Mean (SD) (CI 95%)	*p* K-S	Mean (SD) (CI 95%)	*p* K-S	Mean (SD) (CI95%)	ICC _(1–1)_ (CI 95%)	SEM	CV	MDC	LoA (CI95%)	*p*-Value	*r* (*p*-Value)
Sgarlato	18.02 ± 5.76 (16.74–19.30)	0.001	18.52 ± 6.10 (17.16–19.87)	0.047	18.27 ± 5.83 (16.97–19.57)	0.964 (0.944–0.977)	1.106	3.134	3.066	0.5 (−3.7–4.7)	0.773 **	0.935 ** (<0.001)
Modified Sgarlato	24.13 ± 6.02 (22.79–25.48)	0.200	24.08 ± 6.20 (22.70–25.46)	0.780	24.11 ± 5.93 (22.79–25.43)	0.937 (0.901–0.959)	1.488	4.066	4.126	−0.1 (−5.9–5.8)	0.859 *	0.880 * (<0.001)
Rearfoot	15.27 ± 8.49 (13.38–17.16)	0.036	15.73 ± 8.62 (13.81–17.65)	0.003	15.50 ± 8.47 (13.63–17.37)	0.965 (0.946–0.978)	1.585	1.830	4.392	0.5 (−5.7–6.6)	0.779 **	0.933 ** (<0.001)
Root	15.84 ± 6.31 (14.44–17.25)	0.079	18.21 ± 8.44 (16.33–20.09)	0.200	17.03 ± 7.05 (15.46–18.60)	0.860 (0.728–0.921)	2.638	2.416	7.312	2.4 (−7.1–11.8)	0.092 *	0.824 * (<0.001)
Engel	25.83 ± 5.94 (24.51–27.16)	0.200	24.11 ± 6.3 (22.70–25.53)	0.200	24.97 ± 6.05 (23.62–26.32)	0.950 (0.790–0.980)	1.353	4.127	3.750	−1.7 (−5.9–2.5)	0.060 *	0.942 * (<0.001)
Modified Engel	25.08 ± (6.95) (23.53–26.62)	0.200	23.54 ± 7.08 (21.97–25.12)	0.200	24.31 ± 6.85 (22.79–25.84)	0.941 (0.874–0.968)	1.664	3.549	4.612	−1.5 (−7.4–4.4)	0.171 *	0.908 * (<0.001)
Kite	25.26 ± 7.11 (23.67–26.84)	0.094	27.81 ± 7.93 (26.04–29.57)	0.200	26.53 ± 7.12 (24.95–28.12)	0.854 (0.706–0.919)	2.721	3.726	7.541	2.5 (−7.1–12.2)	0.034 *	0.791 * (<0.001)
Kilmartin	23.20 ± 9.88 (21.00–25.40)	0.140	22.42 ± 9.45 (20.31–24.52)	0.004	22.81 ± 9.55 (20.68–24.93)	0.974 (0.958–0.983)	1.540	2.388	4.268	−0.8 (−6.7–5.1)	0.517 **	0.952 ** (<0.001)
Modified Kilmartin	69.74 ± 5.88 (68.42–71.05)	0.200	67.57 ± 6.36 (66.15–68.99)	0.200	68.65 ± 6.00 (67.32–69.99)	0.928 (0.861–0.972)	1.610	11.442	4.463	−2.2 (−7.0–2.7)	0.032 *	0.921 * (<0.001)
Simons	2.67 ± 9.34 (0.59–4.75)	0.200	2.34 ± 9.89 (0.14–4.54)	0.197	2.51 ± 9.27 (0.44–5.57)	0.924 (0.888–0.951)	2.556	3.693	7.084	0.3 (−9.7–0.4)	0.274 *	0.859 * (<0.001)
Laaveg & Ponseti	−4.22 ± 7.74 (−5.95–−2.50)	0.200	−4.72 ± 8.30 (−6.57–−2.87)	0.200	−4.47 ± 7.83 (−6.21–−2.73)	0.948 (0.919–0.966)	1.786	−1.752	4.949	−0.5 (−7.5 –6.5)	0.196 *	0.903 * (<0.001)

Abbreviations: SD. standard deviation; CI. Confidence Interval; ICC. Intraclass Correlation Coefficient; SEM. Standard Error of Measurement; MCD. Minimum Detectable Change; LoA. 95% limits of agreement; *p* K-S. Kolgomorov Smirnov test and *p* > 0.05 considered normal distribution; r. ** Spearman and * Pearson correlation coefficient; * *p* value from U Mann Whitney; ** *p* value from Independent *t* student test; Statistical significance for a *p* value <0.05. with a 95% confidence interval.

## Data Availability

The datasets used and/or analyzed in the current study or any query regarding to the research process are available from the corresponding author.

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
