# Peer review of "Intra and Inter-Observer Reliability and Repeatability of Metatarsus Adductus Angle in Recreational Football Players: A Concordance Study"

_jcm, 2022, doi:10.3390/jcm11072043_

Round 1
Reviewer 1 Report
The authors must be commended for the large amount of data from their study . The results, discussion and conclusion sections could be improved . The authors would draw new guidelines but the current version needs clarification of the text. May be that improving the English translation readers will better understand the excellent work done by the authors.
Author Response
Response to Reviewer 1 Comments
Point 1:
The authors must be commended for the large amount of data from their study. The results, discussion and conclusion sections could be improved . The authors would draw new guidelines but the current version needs clarification of the text. May be that improving the English translation readers will better understand the excellent work done by the authors.
Answer 1:
The paper has been sent to editing and enclosed is the certificate.

Reviewer 2 Report
I think it is an excellent work, methodologically well done and with interesting iconography. However, since the focus of the study is based on the Intraclass Correlation Coefficient, I would like the authors to specify whether an Absolute Agreement was performed.
On the other hand, I think that the method of recruitment of the subjects should be specified. Was there any randomization?
Finally, in the Limitations of the study, don't you think that the sample size should also be taken into account?
Author Response
Response to Reviewer 2 Comments
Point 1:
I think it is an excellent work, methodologically well done and with interesting iconography. However, since the focus of the study is based on the Intraclass Correlation Coefficient, I would like the authors to specify whether an Absolute Agreement was performed.
Answer 1:
We add: Intraclass Correlation Coefficient two-way random effects, absolute agreement, multiple raters/measurements (ICC 2,k)
Point 2:
On the other hand, I think that the method of recruitment of the subjects should be specified. Was there any randomization?
Answer 2:
In section “subjects” we already have included:
All consecutive recreational football players over 21 years
No, there was no aleatorization due to is a reliability study.
Point 3: Finally, in the Limitations of the study, don't you think that the sample size should also be taken into account?
Answer 3:
In sample size calculation section, we stated:
We calculated the sample size to be 36 patients with a Bonett’s approximation (19)
In results section we added:
A total of 40 participants in each group were recruited, with a total of 80 participants.

Reviewer 3 Report
Dear authors the topic is very interesting. Your methods are well described and the discussion is supported by your results. As regard the introduction (from line 45 to 50), I suggest to underline the importance also of the ct-scan in order to manage the variability of the x-ray measurements.
In fact you may cite the following article were authors used a ct scan evaluation in order to confirm the eficacy of new angular value
“Painful knee prosthesis: CT scan to assess patellar angle and implant malrotation
Muscles, Ligaments and Tendons JournalOpen AccessVolume 6, Issue 4, Pages 461 - 4661 October 2016
Spinarelli A. et al.”
Author Response
Response to Reviewer 3 Comments
Point 1: Dear authors the topic is very interesting. Your methods are well described and the discussion is supported by your results. As regard the introduction (from line 45 to 50), I suggest to underline the importance also of the ct-scan in order to manage the variability of the x-ray measurements.
In fact you may cite the following article were authors used a ct scan evaluation in order to confirm the eficacy of new angular value “Painful knee prosthesis: CT scan to assess patellar angle and implant malrotation Muscles, Ligaments and Tendons JournalOpen AccessVolume 6, Issue 4, Pages 461 - 4661 October 2016 Spinarelli A. et al.”
Answer 1:
We added in limitation section:
Furthermore, computed tomography (CT) scan is a useful diagnostic tool to produce a 3D image of bones and further research is needed to find out reliability and repeatability of metatarsus adductus angles as a new method as postulated by Spinarelli A. et al (47)
